# Whole Yeast Vaccine Displaying ZIKV B and T Cell Epitopes Induces Cellular Immune Responses in the Murine Model

**DOI:** 10.3390/pharmaceutics15071898

**Published:** 2023-07-06

**Authors:** Anna Jéssica Duarte Silva, André Luiz Santos de Jesus, Lígia Rosa Sales Leal, Larissa Silva de Macêdo, Bárbara Rafaela da Silva Barros, Georon Ferreira de Sousa, Simone da Paz Leôncio Alves, Lindomar José Pena, Cristiane Moutinho Lagos de Melo, Antonio Carlos de Freitas

**Affiliations:** 1Laboratory of Molecular Studies and Experimental Therapy—LEMTE, Department of Genetics, Federal University of Pernambuco, Recife 50670-901, Brazil; anna.jessica@ufpe.br (A.J.D.S.); ligiarsleal@gmail.com (L.R.S.L.); larissa.smacedo@ufpe.br (L.S.d.M.); 2Federal Institute of Mato Grosso, Lucas do Rio Verde 78455-000, Brazil; andreluizsjesus@gmail.com; 3Department of Antibiotics, Federal University of Pernambuco, Recife 50670-901, Brazil; barbararsbarros@gmail.com (B.R.d.S.B.); georon.sousa@gmail.com (G.F.d.S.); cristianemout@gmail.com (C.M.L.d.M.); 4Department of Pharmacy, Federal University of Pernambuco, Recife 50670-901, Brazil; simone.leoncio@hotmail.com; 5Department of Virology and Experimental Therapy, Instituto Aggeu Magalhães, Oswaldo Cruz Foundation, Recife 50670-901, Brazil; lindomarvet10@gmail.com

**Keywords:** *P. pastoris*, yeast surface display, multi-epitope, yeast-based vaccines

## Abstract

Improving antigen presentation is crucial for the success of immunization strategies. Yeasts are classically used as biofactories to produce recombinant proteins and are efficient vehicles for antigen delivery, in addition to their adjuvant properties. Despite the absence of epidemic outbreaks, several vaccine approaches continue to be developed for Zika virus infection. The development of these prophylactic strategies is fundamental given the severity of clinical manifestations, mainly due to viral neurotropism. The present study aimed to evaluate in vivo the immune response induced by *P. pastoris* recombinant strains displaying epitopes of the envelope (ENV) and NS1 ZIKV proteins. Intramuscular immunization with heat-attenuated yeast enhanced the secretion of IL-6, TNF-α, and IFN-γ, in addition to the activation of CD4^+^ and CD8^+^ T cells, in BALB/c mice. *P. pastoris * displaying ENV epitopes induced a more robust immune response, increasing immunoglobulin production, especially IgG isotypes. Both proposed vaccines showed the potential to induce immune responses without adverse effects, confirming the safety of administering *P. pastoris* as a vaccine vehicle. Here, we demonstrated, for the first time, the evaluation of a vaccine against ZIKV based on a multiepitope construct using yeast as a delivery system and reinforcing the applicability of *P. pastoris* as a whole-cell vaccine.

## 1. Introduction

Biotechnological-relevant yeast species such as *Saccharomyces cerevisiae*, *Pichia pastoris* (*Komagataella phaffii*), *Hansenula polymorpha*, and *Kluyveromyces lactis* are conventionally employed in the synthesis of immunobiological products [1]. These species have GRAS (Generally Recognized as Safe) status, guaranteeing safety in their application as a biofactory and as a vehicle for vaccine antigens [2,3,4]. The most attractive aspects of this vaccine delivery system are the ability to induce specific immune responses against the recombinant antigen and the yeast adjuvant properties [5]. Whole yeast vaccines can induce higher antigen-specific responses than those promoted by inactivated virus vaccines or proteins conjugated to traditional adjuvants, such as aluminum salts [6,7,8]. 

Yeast-based vaccines have been tested against various infectious agents, including viruses and fungi [9,10,11]. Overall, the recombinant antigens are proteins from the target pathogen or epitopes derived from these proteins that are critical for inducing the appropriate immune response [12,13]. Although immunostimulation associated with the administration of recombinant yeasts can occur regardless of the cellular location of the heterologous protein, the exposure of recombinant antigens on the yeast surface can increase the efficiency of this process [7]. In these systems of yeast surface display, the target protein is covalently linked to an anchor protein with a glycosylphosphatidylinositol (GPI) motif. Anchor proteins belong to the mannoprotein class and include agglutinins (Agα1p, Aga1p, and Aga2p), Flo1p, Sed1p, Cwp1p, Cwp2p, Tip1p, and Tir1p/Srp1p [14,15,16]. The α-agglutinin is the anchor most commonly used and allows the disposition of 10^5^–10^6^ target proteins per cell [17].

Due to the dissemination potential, viral neurotropism, and capacity to induce neurological disorders in fetuses and adults, preventing the Zika virus (ZIKV) infection is considered an issue for global public health [18]. The information about seroprevalence and the duration of immunity against ZIKV is still limited, and the re-emergence of outbreaks continues to be considered [19]. In fact, an increase in the number of infections has been reported in Southeast Asian countries in the last four years [20]. Prophylactic vaccination is one of the main prevention measures, but there are no licensed vaccines for ZIKV infection [21]. Full protection against flaviviruses involves a combination of adaptive humoral and cellular responses [22]. Since many immunodominant epitopes for the induction of T cell-mediated responses are present in non-structural proteins, epitopes or domains of proteins, such as NS1 and NS3, have been included in the design of vaccines, targeting both ZIKV and DENV [23]. In this context, vaccines based on multiepitope sequences are promising platforms concerning immunogenicity, protection, and safety [24].

In a previous study, we developed *P. pastoris* strains harboring multi-epitope antigens derived from the ZIKV envelope (ENV) and NS1 proteins. These strains were evaluated in vitro for their ability to stimulate immune cells and verify the induced response profile, which for some constructs was similar to those promoted by the virus [25]. The main highlights of this study were the stimulus to CD4^+^ and CD8^+^ cell expansion and the secretion of cytokines such as IL-6, IL-10, and TNF-α. This study pointed to the feasibility of using *P. pastoris* as a biotechnological platform for the production of whole yeast vaccines and prompted us to investigate the vaccine responses in vivo through yeast administration in immunocompetent BALB/c mice, evaluating the viability and immunogenicity of this yeast-based vaccine.

## 2. Materials and Methods

### 2.1. Vaccine Antigens and Yeast Strains

Here, we evaluated two yeast recombinant strains named *P. pastoris*:ENV and *P. pastoris*. ENVNS1 was developed in a previous study performed by our group [25]. The vaccine constructs were synthetic multi-epitope antigens composed of epitopes for B and T cells derived from ZIKV ENV and NS1 proteins, predicted in silico. The schematic design of these genes is in Figure 1A. To obtain the recombinant strains, the synthetic genes encoding the multi-epitope constructs were cloned into the expression vector pPGKΔ3_Agα (non-commercial vector; de Almeida et al. 2005) [26] that allows constitutive expression and anchorage of proteins on the yeast surface (Figure 1A). The *P. pastoris* strain transformed with the expression cassettes was GS115 (*his4*; Invitrogen, Waltham, MA, USA). The expression and antigen anchorage were confirmed by RT-PCR and immunofluoresce microscopy, respectively [25]. In addition to these previous analyses, the immunoreactivity and anchoring of the yeasts were evaluated by Yeast-ELISA. For this purpose, 100 µL of 10^7^ cells were applied in 96-well plates for 2 h. Then, the subsequent steps were blocking (5% BSA in PBS) and labeling with an anti-HIS primary antibody (Sigma-Aldrich, St. Louis, MI, USA) 1:1000 diluted (1 h) and a secondary anti-IgG antibody conjugated to peroxidase (Sigma-Aldrich) 1:5000 diluted (45 min). Among the incubations, the samples were triple-washed with PBS-Tween (1%). The revelation was performed with TMB (3,3′,5,5′-tetramethylbenzidine; Life Technologies, Carlsbad, CA, USA) in the dark, and the reaction was interrupted with HCl (1N). The signal was detected on a plate reader with a wavelength set to 450 nm.

### 2.2. Yeast Preparation

After 72 h of cultivation in a YPD medium at 28 °C under agitation (150 rpm), yeast cells were harvested after centrifugation (4500 rpm; 10 min) and washed twice with 1x PBS. Cells were resuspended in sterile 1x PBS to adjust the final concentration to OD600 = 10 in 50 µL. The yeasts were subjected to heat treatment by incubation at 60 °C for 1 h for metabolic inactivation, and stored at 4 °C until the moment of use.

### 2.3. Mice, Ethical Parameters, and Immunization Protocol

Female immunocompetent BALB/c mice, 6–8 weeks old, were raised and maintained in the bioterium of the Aggeu Magalhães Institute (Oswaldo Cruz Foundation—Recife, PE, Brazil) under sterile, pathogen-free conditions. All experiments involving mice strictly followed the standards established by the institutional Ethics Committee for the Use of Animals (protocol No. 110/2017). The immunization schedule was performed in two doses, one week apart, with intramuscular injection (Figure 1B). The mice were divided into three groups (n = 5) defined as G1—mice inoculated with non-recombinant *P. pastoris* (nr); G2—mice vaccinated with *P. pastoris*:ENV; and G3—mice vaccinated with *P. pastoris*:ENVNS1. Before each immunization, all mice were anesthetized with xylazine hydrochloride (10 mg·Kg^−1^) and ketamine (115 mg·Kg^−1^). Each experimental group received doses of 50 µL with yeast cells in an OD_600_ = 10. Twenty-one days after the first dose, all animals were anesthetized for blood collection and subsequent euthanasia. Spleens were removed for isolation and culture of lymphocytes.

### 2.4. In Vitro Culture and Stimulation of Isolated Spleen Lymphocytes

Splenocytes from vaccinated animals were isolated and the mononuclear immune cells were isolated by separation with Ficoll-Paque PLUS 1.077 g·mL^−1^ (GE Healthcare Life Sciences, Uppsala, Sweden) and distributed in 48-well plates, 10^6^ cells/well. The isolated cells were restimulated in vitro with the yeasts *P. pastoris*:nr (for G1), *P. pastoris* expressing ENV epitopes (for G2), and ENVNS1 epitopes (for G3), at a concentration of 10^5^ cells/well. The cells were incubated in an RPMI medium (Sigma-Aldrich) containing 10% FBS at 37 °C (5% CO_2_), at the experimental times of 24 h, 48 h, and 72 h.

### 2.5. Immunological Analysis

Lymphocytes isolated from blood and the spleen were characterized for the presence of surface markers CD4, CD8, and CD16 by labeling the cells with the corresponding antibodies (anti-CD4-FITC, anti-CD8-PE, and anti-CD16/32-FITC; BD^TM^ Bioscience, Franklin Lakes, NJ, USA). Serum and culture supernatant were analyzed for cytokine dosage of TNF-α, IFN-γ, IL-2, IL-4, IL-6, IL-10, and IL-17A using the BD CBA Mouse Th1/Th2/Th17 Kit (BD^TM^ Bioscience), following the manufacturer’s instructions. IgG, IgM, IgA, and IgE immunoglobulins were measured in the serum of vaccinated mice using the Mouse Immunoglobulin Isotyping Kit (BD^TM^ Bioscience). All acquisitions for immunological assays were performed by flow cytometry (BD *ACCURI C6*).

### 2.6. Hematological and Biochemical Analyses

Blood samples were collected using a cardiac puncture, placed into EDTA-K2 tubes, and centrifuged at 3600 rpm for 10 min to separate serum and plasma. The hematological evaluation included global counts of red blood cells, leukocytes, platelets, determination of hematocrit, and hemoglobin concentration. The values of red blood cells, hematocrit, and hemoglobin allowed the calculation of the mean corpuscular volume and the mean corpuscular hemoglobin concentration. The blood cell count was performed in a Neubauer chamber, with differential counting of the slides stained by the Rapid Panoptic method [27]. The platelet number was determined using the Fonio method [28]. Hemoglobin was measured by colorimetry using a spectrophotometer. The biochemical analyses were performed with Labtest Diagnostic kits (Lagoa Santa, MG, Brazil). The levels of glucose, urea, and creatinine were measured with end-point colorimetric enzymatic assays. Alkaline phosphatase was detected by a modified Roy’s method, and the liver transaminases by the Reitman–Frankel method.

### 2.7. Statistical Analysis

Graphs and statistical analyses were generated by GraphPad Prism version 7.04. The analysis of variance (ANOVA) was applied to assess statistical differences between groups. Results with a *p*-value < 0.05 were considered statistically significant.

## 3. Results

### 3.1. P. pastoris Can Surface Display Vaccine Antigens

The recombinant yeasts were generated and characterized in a previous study [25]. A Yeast-ELISA was performed to confirm the accessibility of anchored antigens. *P. pastoris*:ENV and *P. pastoris*:ENVNS1 showed higher immunoreactivity compared to non-recombinant yeast (NR), confirming protein anchoring (Figure 2).

### 3.2. Recombinant P. pastoris Strains Induce Increased Secretion of Serum and Splenic Cytokines

The vaccine constructs promoted different cytokine patterns. Regarding serum cytokines, *P. pastoris*:ENV caused an increase in the levels of IL-2, TNF-α, and IL-17A. On the other hand, *P. pastoris*:ENVNS1 increased all analyzed cytokines, overcoming both non-recombinant yeast and *P. pastoris*:ENV, mainly in IL-2, IL-4, IL-10, and IL-1A (Figure 3). The response elicited by both recombinant yeasts was equivalent only to the TNF-α release. The profile of cytokines secreted by splenic lymphocytes, restimulated in vitro, was more restricted than observed in animal serum, with a greater tendency toward a Th1 pattern. *P. pastoris*:ENV caused an increase in IL-6 (48 h) and IFN-γ at 24 h and 48 h (Figure 4A,B), while *P. pastoris*:ENVNS1 induced IL-6 and TNF-α in 72 h cultures (Figure 4A–C). There was no significant production of the other cytokines (Appendix A). Non-recombinant yeast did not cause a significant stimulus compared to *P. pastoris* expressing vaccine antigens.

### 3.3. P. pastoris:ENV and P. pastoris:ENVNS1 Enhance Antibody Production

The levels of different types and isotypes of immunoglobulins were assessed to evaluate the humoral response. Overall, the *P. pastoris*:ENV vaccine stood out compared to ENVNS1 and non-recombinant yeast (Figure 5A–D). *P. pastoris*:ENV induced production of all of the IgG isotypes, especially IgG3. Both vaccine constructs elicited a similar profile for IgA and IgM (Figure 5E,G), superior to *P. pastoris*:nr. The two vaccines led to an increase in IgE, mainly *P. pastoris*:ENV (Figure 5F). The ratio of IgG2a/IgG1 indicates the response profile type (Th1 or Th2) and was similar for all three yeast strains. Although the mean for *P. pastoris*:nr (1.27) was higher than the recombinants (1.14 and 1.03), the difference was not statistically significant (Figure 5H). Values greater than 1 indicate a tendency toward a Th1 response pattern. However, the proximity between the averages of IgG1 and IgG2a for all groups suggested a balance between Th1 and Th2 responses. 

### 3.4. Recombinant Yeasts Stimulate Cellular Responses

The evaluation of the cellular response induced by vaccination was based on the analysis of CD4^+^, CD8^+^, and CD16^+^ T lymphocyte populations present in the blood and spleen after immunization. Regarding circulating lymphocytes, mice inoculated with non-recombinant *P. pastoris* promoted a higher stimulus to CD4^+^ T lymphocytes. No difference in the number of CD4^+^ T cells between the recombinant yeasts was observed (Figure 6A). On the other hand, regarding the number of CD8^+^ T cells, *P. pastoris*:nr and *P. pastoris*:ENVNS1 showed a similar pattern, superior to that induced by *P. pastoris*:ENV (Figure 6B). To Natural Killer (NK) CD16^+^ T lymphocytes, *P. pastoris*:ENV and *P. pastoris*:ENVNS1 promoted an increase in the number of cells compared to the non-recombinant (Figure 6C). 

Lymphocytes isolated from the spleen of the immunized animals were restimulated with the respective strain and incubated for periods of 24 h, 48 h, and 72 h. The most significant stimulus was observed for the CD4^+^ T lymphocyte subpopulation. The immunization with the recombinant yeasts, expressing ENV and ENVNS1, induced an increase in the amount of CD4^+^ T cells, especially when observing the 72 h culture, where both were superior to non-recombinant *P. pastoris* (Figure 7A). The levels of CD8^+^ T cells were lower than the expression of CD4^+^ and CD16^+^ and substantially reduced throughout the culture period. The proportion of CD8^+^ T cells stimulated by *P. pastoris*:ENV stood out at times of 24 h and 48 h. After 72 h of stimulus, the culture plates of splenocytes from animals inoculated with *P. pastoris*:ENVNS1 had a higher number of CD8^+^ T lymphocytes (Figure 7B). Regarding the CD16^+^ T cell profile, *P. pastoris*:ENV induced a significant increase in the first 24 h; however, it declined at 48 h and 72 h. Meanwhile, *P. pastoris*:ENVNS1 caused a more significant expansion at 48 h and 72 h, even though it also diminished at 72 h (Figure 7C).

### 3.5. The Whole Yeast Vaccines Do Not Cause Significant Side Effects

Animals were weighed on the first day of the immunization regimen, on the booster dose day, and 21 days after the initial dose (Appendix A). Throughout the vaccine schedule, there was no weight loss or significant behavioral changes. There were no points of inflammation, swellings, or adverse effect in the injection site. Overall, there were no biochemical alterations that would qualify as clinical disturbances (Table 1). The hematological changes were relative to leukocytosis and lymphopenia, in addition to neutrophilia in the three experimental groups.

## 4. Discussion

In this study, we evaluated a multiepitope vaccine based on B and T cell epitopes from Env and NS1 ZIKV proteins in a yeast surface display strategy. The whole yeast vaccines evaluated were developed in a previous study that assessed, in vitro, the potential of *P. pastoris* yeast as an adjuvant and vaccine platform [25]. The immune analyses showed the induction of a Th1-type immune response based on the profile of cytokines (IL-6, TNF-α) and activated T cells (CD4^+^ and CD8^+^). Interestingly, in this same study, the construction *P. pastoris*:ENVNS1 showed the induction of an immunological profile similar to that promoted by the virus when incubated with splenocytes from BALB/c mice. Together, the results enabled the continuation of the evaluation using a pre-clinical test in a murine model. 

After a two-dose schedule, the immune response was assessed by cytokine dosage, immunophenotyping, and immunoglobulin production. BALB/c mice immunized with recombinant *P. pastoris* showed a serum elevation of IL-2 and TNF-α for the two tested vaccines, in addition to the induction of IL-17A. Noteably, in addition to its conventional regulatory role, IL-17A may play pro-inflammatory functions in some viral, fungal, and cancer infections [32]. Moreover, high levels of IFN-γ, TNF-α, and IL-2 secreted by CD4^+^ and CD8^+^ T lymphocytes are also detected in immunocompetent C57BL/6 mice infected with ZIKV, exhibiting a Th1 pattern [33,34]. 

Despite eliciting Th1 cytokines, the *P. pastoris*:ENVNS1 vaccine induced an increase in IL-4 and IL-10, which may be a consequence of follicular T cell (Thf) activation. In Zika virus infection, Tfh cells enhanced the production of cytokines, such as IL-4 and IL-21, and also acted in a Th1-like manner, producing IFN-γ. Additionally, this lymphocyte subpopulation influences the development of neutralizing antibodies against ZIKV [35]. The detection of cytokines such as IL-2, IL-4, and IL-17 in individuals in the acute phase of ZIKV infection suggests a polyfunctional response profile characterized by Th1, Th2, and Th17 responses [36]. Concerning the splenic immune cells, the cytokine dosage indicated a Th1 profile, where *P. pastoris*:ENV induced an increase in IL-6 and IFN-γ levels, while *P. pastoris*:ENVNS1 stimulated IL-6 and TNF-α production. These pro-inflammatory cytokines compose the anti-ZIKV immune response and were also observed, in vitro, in assays performed with these recombinant yeasts in previous studies [25,37]. The role of IL-6 resulting from the administration of whole yeast vaccines has been linked to the generation of CD4^+^ T cells [38]. Furthermore, increased production of TNF-α and IFN-γ is related to the activation of CD4^+^ and CD8^+^ T cells that mediate effective responses against ZIKV-infected cells [39].

Regarding circulating lymphocytes, the recombinant antigens seem to modulate the CD8^+^ T cells since there was a significant difference between the vaccines. Non-recombinant *P. pastoris* and expressing ENVNS1 induced a greater stimulus than *P. pastoris*:ENV. The NS1 epitopes may have influenced the activation of the CD8^+^ T lymphocytes, which reinforces the importance of including immunodominant epitopes present in non-structural proteins in the design of vaccines for flaviviruses [39]. In addition, both recombinant yeasts elicited the expansion in the number of CD16^+^ T lymphocytes, mainly *P. pastoris*:ENVNS1 construction.

Analysis of splenic lymphocytes reflects the induction of immune memory responses [40]. The stimulus promoted by recombinant yeasts was greater for the CD4^+^ T cell population than CD8^+^ T cells, overall. *P. pastoris*:ENV stimulated CD4^+^ (48 h and 72 h), CD8^+^ (24 h and 48 h), and CD16^+^ (24 h and 72 h) cells. *P. pastoris*:ENVNS1 elicited an increase in the number of CD4^+^ cells (24 h and 72 h), CD8^+^ (72 h), even in low levels, and CD16^+^ (48 h and 72 h). The vaccine responses reported are in line with other studies about the importance of T cells in the context of vaccine development, indicating that a coordinated balance among the action of antibodies, Tfh CD4^+^, Th1 CD4^+^, and CD8^+^ T cells are essential to the infection control and long-term protection [39,41]. 

In addition to the cellular immune response, the recombinant yeasts induced significant production of different classes of immunoglobulins. The highest levels were observed for IgG3, IgA, and IgM. The *P. pastoris*:ENV vaccine promoted a significant increase in the production of all IgG isotypes. Elevations in the generation of IgG1 and IgG2a were observed in mice challenged with ZIKV PE243 [42] and immunized with VLPs [43]. The IgG2 isotype is predominant among neutralizing antibodies that induce protection and is important for ZIKV infection clearance [35,42]. Additionally, there were elevations in IgG3 and IgM, commonly observed in the acute phase of viral infections, which were involved in pro-inflammatory effector mechanisms [43,44].

Both recombinant yeasts caused an increase in IgE. Antiviral functions performed by IgE have been investigated, but there is no consistent data regarding the role of this immunoglobulin in ZIKV infection [45]. Once there were no changes in eosinophil levels or signs of anaphylaxis, the contribution of IgE was possibly related to non-allergic defense mechanisms induced by the vaccine. The two vaccine constructs also promoted an increase in IgA and IgM at equivalent levels. These immunoglobulins are frequent in recent ZIKV infections, making them helpful for diagnostic assays [46].

All tested yeasts induced neutrophilia in inoculated mice. In addition to acting as critical cells of innate immunity, neutrophils also contribute to adaptive immunity in the transport and presentation of antigens and the regulation of antigen-specific responses [47]. Although neutrophil activation is involved in the pathogenesis of ZIKV infection, the role of these cells in the antiviral immune response generated by vaccination is unclear [48]. 

Overall, the results obtained suggest the involvement of Th1 and Th2 responses, providing a polyfunctional immune response profile also observed in the ZIKV infection [33]. Similar patterns are observed in vaccine studies for ZIKV based on different platforms, such as EDIII-based subunit vaccines [49] and VLPs [50]. This study explored the biotechnological potential of *P. pastoris* regarding its applicability as a biofactory and antigen carrier. We observed that although previous studies show yeast immunostimulatory properties, the expression of vaccine antigens can modulate the immune response. Recombinant *P. pastoris* promoted an increase in immunoglobulins production and cellular immune response activation. There were no deaths, diarrhea, or weight loss in the animals vaccinated with yeasts during the vaccination schedule, which was in agreement with previous studies using *P. pastoris* and other species [51]. 

In vivo evaluations of whole yeast vaccines are essential to validate this biotechnological platform highlighted as an alternative vaccine antigen delivery system. The biotechnological potential of *P. pastoris* was explored in different ways: as a biofactory, as an antigen carrier, and as a vaccine adjuvant, showing its versatility. *P. pastoris* is pointed out as a cost-effective platform for vaccine production, favoring production and distribution in low- and middle-income countries [52]. 

## 5. Conclusions

Despite the several immunoinformatic-based studies for predicting ZIKV epitopes, data at the preclinical level are still scarce. This study demonstrates, for the first time, the evaluation of a multi-epitope construct as a vaccine antigen for ZIKV using yeast as a vehicle with the surface display system. As the main product, *P. pastoris*:ENV proved to be a promising vaccine candidate, inducing the activation of T CD4^+^ and T CD8^+^ lymph;ocytes and effector cytokines, as well as producing immunoglobulins, compared to the other experimental groups. Together, the results point to the generation of immune responses that should be tested to verify the induction of protection against ZIKV infection. 

## Figures and Tables

**Figure 1 pharmaceutics-15-01898-f001:**
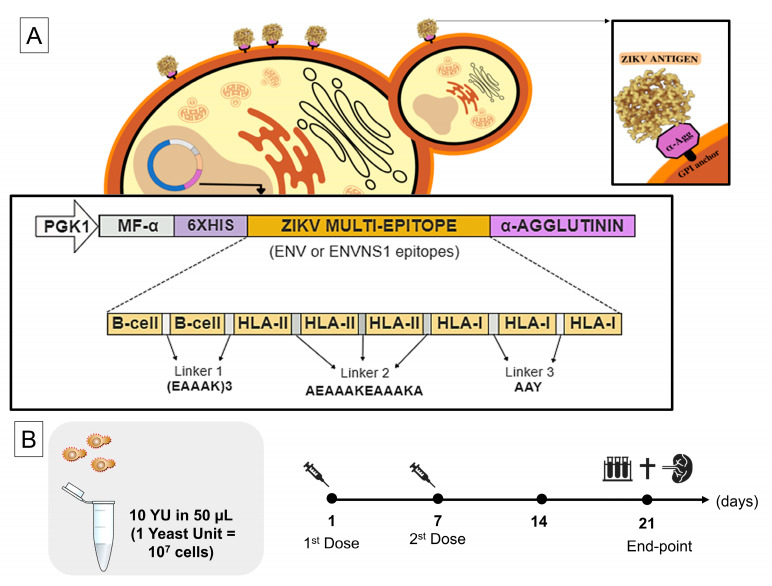
Experimental design. (**A**) The first phase of the study included the selection of immunogenic epitopes from the envelope and NS1 proteins, the construction of the expression cassettes, and the obtaining of the recombinant yeasts [25]. The sets of Env and Env + NS1 epitopes were fused to the α-agglutinin anchor protein, allowing the display of the recombinant proteins on the *P. pastoris* surface. PGK1: promoter; MF-α: signal peptide for protein secretion; 6H: 6x His-tag for immunodetection. (**B**) The concentration of yeasts cells was 10 YU, and these cells were heat-inactivated before immunization procedures. The BALB/c mice received two doses of each yeast preparation, via intramuscular, and were monitored for weight and activity throughout the 21 days of the experiment. Blood collection, euthanasia, and removal of the spleen were performed on the last day of the schedule.

**Figure 2 pharmaceutics-15-01898-f002:**
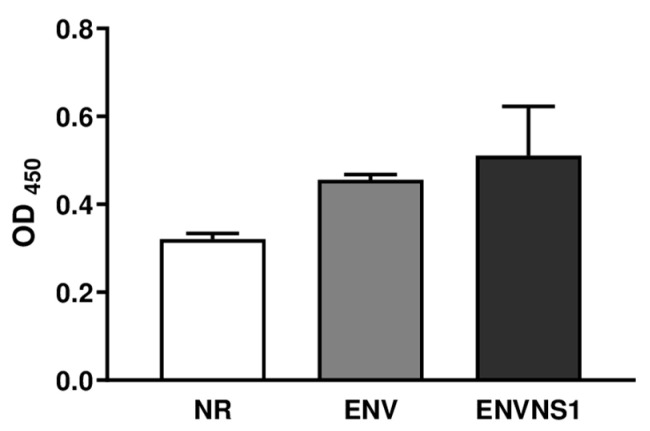
Yeast-ELISA was performed to evaluate the protein expression and anchorage of the antigens on *P*. *pastoris* surface. These yeast lineages were utilized in the immunization assay.

**Figure 3 pharmaceutics-15-01898-f003:**
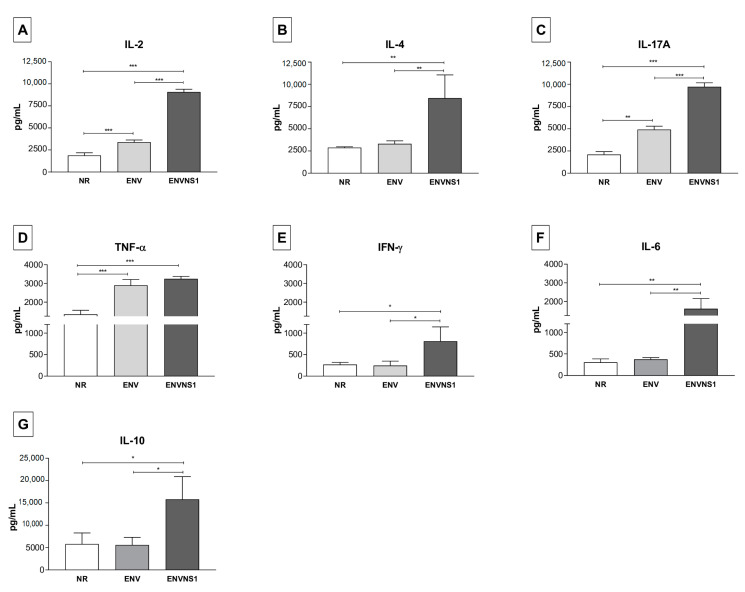
Serum dosage of cytokines from the blood collected 14 days after the second dose. (**A**) IL-2, (**B**) IL-4, (**C**) IL-17A, (**D**) TNF-α, (**E**) IFN-γ, (**F**) IL-6, (**G**) IL-10. *P. pastoris*:ENVNS1 stood out concerning the production of all tested cytokines. Cytokine values were measured in pg·mL^−1^. Asterisks represent statistical significance (* *p* < 0.05, ** *p* < 0.01, *** *p <* 0.001). Bar = means ± SD.

**Figure 4 pharmaceutics-15-01898-f004:**
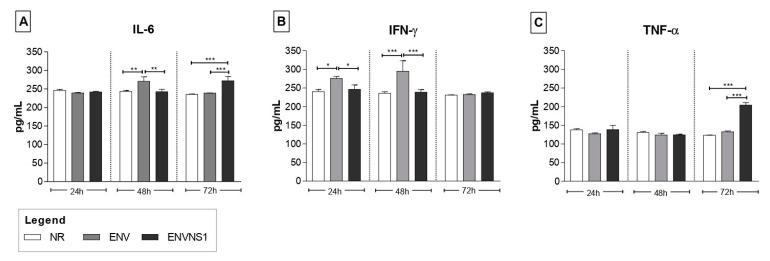
Analysis of cytokine levels in the supernatant of the cultures of mononuclear immune cells, isolated from the spleens of vaccinated animals, after in vitro restimulation. The culture periods analyzed were 24 h, 48 h, and 72 h. (**A**) IL-6, (**B**) IFN-γ, (**C**) TNF-α. Cytokine values were measured in pg·mL^−1^. Asterisks represent statistical significance (* *p* < 0.05, ** *p* < 0.01, *** *p <* 0.001). Bar = means ± SD.

**Figure 5 pharmaceutics-15-01898-f005:**
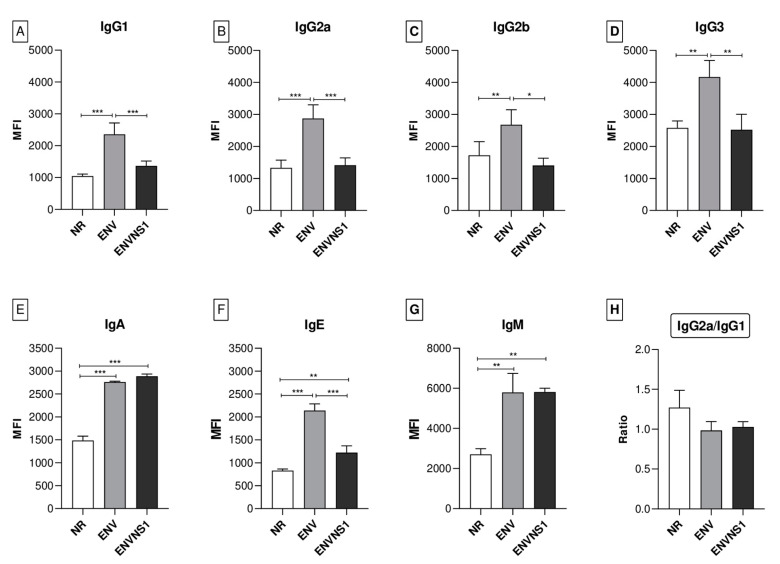
Secretion profile of different immunoglobulin isotypes in mouse serum 14 days after the second dose. The immunoglobulins IgG1 (**A**), IgG2a (**B**), IgG2b (**C**), IgG3 (**D**), IgA (**E**), IgE (**F**), and IgM (**G**) were detected by flow cytometry. The IgG2a/IgG1 ratio was calculated for the three experimental groups (**H**). The values in MFI correspond to the mean fluorescence intensity. Asterisks represent statistical significance (* *p* < 0.05, ** *p* < 0.01, *** *p* < 0.001). Bar = means ± SD.

**Figure 6 pharmaceutics-15-01898-f006:**
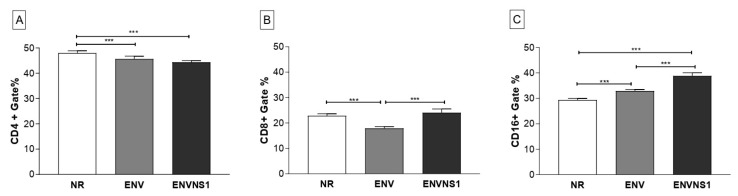
Cellular immune response. Expression of surface markers in blood lymphocytes after immunization. (**A**) % of cells CD4^+^, (**B**) % of cells CD8^+^, (**C**) % of cells CD16^+^. Asterisks represent statistical significance (*** *p <* 0.001). Bar = means ± SD.

**Figure 7 pharmaceutics-15-01898-f007:**
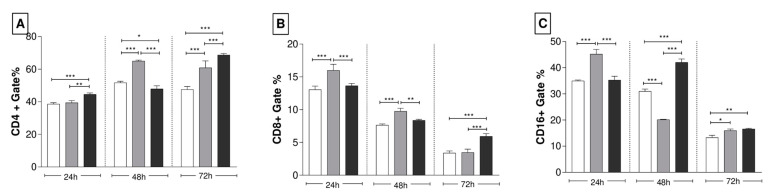
Expression profile of splenic lymphocyte surface markers. These cells were restimulated in vitro and cultured for the 24 h, 48 h, and 72 h experimental periods. (**A**) % of cells CD4^+^, (**B**) % of cells CD8^+^, (**C**) % of cells CD16+. Asterisks represent statistical significance (* *p* < 0.05, ** *p* < 0.01, *** *p <* 0.001). Bar = means ± SD.

**Table 1 pharmaceutics-15-01898-t001:** Hematological and biochemical parameters of vaccinated mice. Values correspond to mean ± standard deviation.

Analysis	NR	ENV	ENVNS1	R.V. [29,30,31]
Hematological				
Red blood cells (10^6^/mm^3^)	5.27 ± 0.35	5.32 ± 0.49	4.7 3± 0.41	7.3 ± 2.01
Hemoglobin (g·dL^−1^)	15.18 ± 1.10	16.04 ± 1.89	14.05 ± 1.31	13.82 ± 1.07
Hematocrit (%)	47.4 ± 3.36	48.2 ± 5.67	42.25 ± 3.86	38.44 ± 3,93
MCV (fL)	89.82 ± 1.27	90.35 ± 2.43	89.24 ± 0.62	60.26 ± 18.25
MCHC (%)	33.29 ± 0.06	33.27 ± 0.05	33.27 ± 0.08	33.00 ± 2.60
Total leukocytes (10^3^/mm^3^)	9.38 ± 1.14	9.16 ± 0.58	9.2 ± 0.42	6.23 ± 2.57
Neutrophils (%)	38.8 ± 5.72	41 ± 6.20	44.25 ± 2.93	22.96 ± 5.54
Lymphocytes (%)	56 ± 3.16	56.6 ± 6.07	53.5 ± 2.52	71.76 ± 5.9
Eosinophils (%)	1.6 ± 0.89	1.2 ± 0.45	1 ± 0	2.16 ± 1.71
Monocytes (%)	1.6 ± 0.89	1.2 ± 0.45	1.25 ± 0.5	2.68 ± 1
Platelets (10^3^/mm^3^)	386 ± 39.06	459.2 ± 11.73	434 ± 37.21	560 ± 119
Biochemical tests				
Glucose (mg·dL^−1^)	71.9 ± 4.34	85.84 ± 5.95	80.57 ± 13.86	80.75 ± 20.25
AST (UI·L^−1^)	133.52 ± 5.77	143.8 ± 0.08	103.02 ± 25.98	239.50 ± 141.20
ALT (UI·L^−1^)	154.4 ± 6.57	145.32 ± 5.11	145.67 ± 17.23	156.70 ± 57.20
ALP (UI·L^−1^)	215.82 ± 8.88	218.16 ± 4.52	202.6 ± 22.93	362.90 ± 226.60

RV—Reference Value; MCV—Mean Corpuscular Volume; MCHC—Mean Corpuscular Hemoglobin Concentration; AST—Aspartate Aminotransferase; ALT—Alanine Aminotransferase; ALP—Alkaline Phosphatase.

## Data Availability

Data sharing is not applicable to this article.

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
