# Peer review of "Whole Yeast Vaccine Displaying ZIKV B and T Cell Epitopes Induces Cellular Immune Responses in the Murine Model"

_pharmaceutics, 2023, doi:10.3390/pharmaceutics15071898_

Round 1
Reviewer 1 Report
In this manuscript, Silva et al. perform the induction of an antigen-specific immune response in mice using antigen-expressing yeast as a vehicle. Although in principle of interest to the field, the manuscript requires significant revision in many aspects, including clarity of presentation as well as conclusions drawn: this means a more detailed description of the methods used, the results obtained and a more extended discussion.
- Materials & methods, Results & Discussion, Legends:
- Figure 1: the presentation of the epitopes in B is very small and makes it difficult to understand. C: the "t" at "recombinant" is displaced. How are “10 YU” defined?
- Are 3.1. and Fig. 2 really necessary? The main (and only) message is “successful protein expression”.
- How did the authors ensure that the different yeast transformants expressed comparable amounts of antigen?
- Why did the authors choose a rather unusual site of application (left tibial muscle)?
- Why were CD8 T cells from serum and not from spleen examined?
- Typing error in line Fig.2: DO instead of OD.
- Why was whole yeast used for restimulation and not the pure epitope? How do the authors distinguish between yeast- and epitope-specific responses? How do the authors explain the differences between the epitope-expressing yeast? Why are NK cells addressed that per se do not belong to the adaptive immune response?
- What about a more detailed comparison to the authors own already published in vitro results given in [25]?
- Figure captions suffer, at least in part, from inadequate description of the methods used as well as adequate description of the legends shown in the figure. Caption to Figure 5 contains no explanation of A, B, C… although these numbers are in the figure.
The manuscript contains several typos that should be removed.
Author Response
Responses to Reviewers
Reviewer 1
- Figure 1: the presentation of the epitopes in B is very small and makes it difficult to understand. C: the "t" at "recombinant" is displaced. How are “10 YU” defined?
Answer: We have restructured Figure 1 to highlight the surface display system, the expression cassette, and the multi-epitope antigen. YU corresponds to “Yeast Unit” and 1 YU = 107 cells (data added in the figure).
- Are 3.1. and Fig. 2 really necessary? The main (and only) message is “successful protein expression”.
Answer: We appreciate your observation. However, we believe it is interesting to keep the result of the yeast-ELISA once it not only shows the successful protein expression but also points to the successful anchorage of the proteins on the surface of the yeast (in addition to the immunofluorescence microscopy, published in the previous study – Silva et al. 2021).
- How did the authors ensure that the different yeast transformants expressed comparable amounts of antigen?
Answer: From the results observed in the Yeast-ELISA, it is possible to infer that the amount of protein anchored in the recombinant strains is similar, and the signal of both is higher than that observed in the non-recombinant yeast (as expected).
- Why did the authors choose a rather unusual site of application (left tibial muscle)?
Answer: We believe that there may have been some error in this placement, as this site was not mentioned in the text of the methodology, as follows:
“Female immunocompetent BALB/c mice, 6 – 8 weeks old, were raised and maintained in the bioterium of the Aggeu Magalhães Institute (Oswaldo Cruz Foundation – PE, Brazil) under sterile, pathogen-free conditions. All experiments involving mice followed strictly the standards established by the institutional Ethics Committee for the Use of Animals (protocol n. 110/2017). The immunization schedule was performed in two doses, one week apart, with intramuscular injection (Figure 1C). The mice were divided into three groups (n=5) defined as: G1 - mice inoculated with non-recombinant P. pastoris (nr); G2 - mice vaccinated with P. pastoris:ENV; and G3 - vaccinated with P. pastoris:ENVNS1. Before each immunization, all mice were anesthetized with xylazine hydrochloride (10 mg.Kg-1) and ketamine (115 mg.Kg-1). Each experimental group received doses of 50 µl with yeast cells in an OD600=10. Twenty-one days after the first dose, all animals were anesthetized for blood collection and subsequent euthanasia. Spleens were removed for isolation and culture of splenic lymphocytes.”
- Why were CD8 T cells from serum and not from spleen examined?
Answer: As shown in topic 3.4 (Recombinant yeasts stimulate cellular responses) we analyzed CD8 T cells from both blood (Figure 6B) and spleen (Figure 7B).
- Typing error in line Fig.2: DO instead of OD.
Answer: This error was corrected in the text.
- Why was whole yeast used for restimulation and not the pure epitope? How do the authors distinguish between yeast- and epitope-specific responses? How do the authors explain the differences between the epitope-expressing yeast? Why are NK cells addressed that per se do not belong to the adaptive immune response?
Answer: Once again I appreciate the referee's attention, however, although the approach presented by him is correct, this fact does not invalidate our approach. The restimulation carried out with the respective yeasts of each experimental group also allows the evaluation of the immunological impact of each construction. The differences between the negative control (non-recombinant yeast) and the tested vaccines indicate the specificity of the results. Although NK cells have more direct participation in the innate antiviral response, some studies (see references below) point out that these cells can undergo expansion at the beginning of the infection, detect ZIKV (via NKG2D/MIC-A/B) and induce the production of pro-viral cytokines, which together contribute to adaptive anti-ZIKV responses. We also carried out this analysis in the previous study (see reference below) that was carried out in vitro to understand the dynamics of these cells through stimuli in culture or after the immunization process.
- Kujur, W.; Murillo, O.; Adduri, R.S.R.; Vankayalapati, R.; Konduru, N.V.; Mulik, S. Memory like NK Cells Display Stem Cell like Properties after Zika Virus Infection. PLoS Pathog 2020, 16, e1009132, doi:10.1371/journal.ppat.1009132.
- Maucourant, C.; Nonato Queiroz, G.A.; Corneau, A.; Leandro Gois, L.; Meghraoui-Kheddar, A.; Tarantino, N.; Bandeira, A.C.; Samri, A.; Blanc, C.; Yssel, H.; et al. NK Cell Responses in Zika Virus Infection Are Biased towards Cytokine-Mediated Effector Functions. The Journal of Immunology 2021, 207, 1333–1343, doi:10.4049/jimmunol.2001180.
- Rodrigues De Sousa, J.; Azevedo, R.D.S.D.S.; Quaresma, J.A.S.; Vasconcelos, P.F.D.C. The Innate Immune Response in Zika Virus Infection. Reviews in Medical Virology 2021, 31, doi:10.1002/rmv.2166.
- Silva, A.J.D.; Jesus, A.L.S.; Leal, L.R.S.; Silva, G.A.S.; Melo, C.M.L.; Freitas, A.C. Pichia Pastoris Displaying ZIKV Protein Epitopes from the Envelope and NS1 Induce in Vitro Immune Activation. Vaccine 2021, 39, 2545–2554, doi:10.1016/j.vaccine.2021.03.065.
- What about a more detailed comparison to the authors own already published in vitro results given in [25]?
Answer: We added some more information on the already published data in the article discussion.
- Figure captions suffer, at least in part, from inadequate description of the methods used as well as adequate description of the legends shown in the figure. Caption to Figure 5 contains no explanation of A, B, C… although these numbers are in the figure.
Answer: We appreciate your observation, and all captions have been revised.

Reviewer 2 Report
Novel recombinant ZIKV vaccines are designed based on Pichia pastoris. The research results mainly focus on cellular immunity, which show the potential application of the candidate vaccines to a certain extent, but humoral immunity was not well done. And there is a flaw that the animal model for the evaluation of ZIKV challenge protection has not been established.
Main comments:
1.The reason why humoral immunity such as neutralizing antibody was not done should be added in discussion.
2.The reason why animal challenge was not done should be added in discussion.
3.Conclusions were too long, and it should be shorten to no more than two sentences.
Some minor comments should be revised as follows:
Line 117. Correct “OD600 10” to “OD600=10”.
Line 214. IgA, IgE, IgM and other antibodies were detected, and it was concluded that P. pastoris:ENV and P. pastoris:ENVNS1 enhanced antibody production. It is not sufficient because specific neutralizing antibodies are the most direct and important indicator to evaluate candidate vaccines, so it is recommended to supplement corresponding experiments.
Line 226. The ordinate in Figure 5H should be ratios, not MFI.
Line 227. The data in Figure 5A-D was not descried in the “Results”.
Line 252-254. “After 72 hours of stimulus, the animals inoculated with P. pastoris: ENVNS1 had a higher number of CD8+ T lymphocytes.” This is a wrong statement. Animals cannot accept 72h stimulation, only for different groups of spleen lymphocytes.
Line 259. The three graphs in Figure 7 do not need to maintain the same ordinate scale, which can cause significant differences to be unclear.
Line 262. Here, only the safety of candidate vaccines was evaluated after the assessment of immune indicators. Animal models of Zika virus infection and challenge tests were not established to further verify the efficacy of the candidate vaccines.
Line 349. As the conclusion of the paper, the authors do not seem to summarize the results, specifically which of the two vaccines had the better immune effects.
The manuscript should be revised by an English native speaker. There are many expressions or grammatical errors in the language.
Author Response
Responses to Reviewer 2
Main comments:
1.The reason why humoral immunity such as neutralizing antibody was not done should be added in discussion.
Answer: We appreciate the suggestion regarding the neutralization assays, and we consider it one of the major parameters in the prophylactic vaccine studies. However, the main focus of the submitted manuscript was present Pichia pastoris as a biotechnological platform to deliver vaccine antigens. Thus, the performed assays allowed the evaluation of the immunogenicity of the vaccine constructs, both concerning the vaccine platform (yeast cells) and the carried antigen (derived from the ZIKV).
2.The reason why animal challenge was not done should be added in discussion.
Answer: We agree with the importance of including animal challenges to assess the induction of protection. Nonetheless, the experimental design adopted in the submitted article is similar to that of other studies that aimed to evaluate vaccine candidates against ZIKV infection. Even without performing the immunological challenge, our study points out the immunogenic potential of recombinant yeasts, showing a response pattern close to the expected in the context of natural ZIKV infection and through vaccination (Cibulski et al. 2017; Yang et al. 2017).
- Cibulski, S., Varela, A.P.M., Teixeira, T.F., Cancela, M.P., Sesterheim, P., Souza, D.O., Roehe, P.M., and Silveira, F. (2021). Zika Virus Envelope Domain III Recombinant Protein Delivered With Saponin-Based Nanoadjuvant From Quillaja brasiliensis Enhances Anti-Zika Immune Responses, Including Neutralizing Antibodies and Splenocyte Proliferation. Front. Immunol. 12, 632714.
- Yang, M., Dent, M., Lai, H., Sun, H., and Chen, Q. (2017). Immunization of Zika virus envelope protein domain III induces specific and neutralizing immune responses against Zika virus. Vaccine 35, 4287–4294.
3.Conclusions were too long, and it should be shorten to no more than two sentences.
Answer: We revised the text of the conclusion to make it more direct.
Some minor comments should be revised as follows:
- Line 117. Correct “OD600 10” to “OD600=10”.
Answer: This error has been corrected in the text.
- Line 214. IgA, IgE, IgM and other antibodies were detected, and it was concluded that P. pastoris:ENV and P. pastoris:ENVNS1 enhanced antibody production. It is not sufficient because specific neutralizing antibodies are the most direct and important indicator to evaluate candidate vaccines, so it is recommended to supplement corresponding experiments.
- Line 226. The ordinate in Figure 5H should be ratios, not MFI.
Answer: This has been corrected in the Figure.
- Line 227. The data in Figure 5A-D was not descried in the “Results”.
Answer: The text was adjusted.
- Line 252-254. “After 72 hours of stimulus, the animals inoculated with P. pastoris: ENVNS1 had a higher number of CD8+ T lymphocytes.” This is a wrong statement. Animals cannot accept 72h stimulation, only for different groups of spleen lymphocytes.
Answer: We have corrected this information in the text: “After 72 hours of stimulus, the culture plates of splenocytes from animals inoculated with P. pastoris:ENVNS1 had a higher number of CD8+ T lymphocytes (Figure 7B)”.
- Line 259. The three graphs in Figure 7 do not need to maintain the same ordinate scale, which can cause significant differences to be unclear.
Answer: We appreciate your observation, and we have adjusted the scale of the graphs to facilitate the visualization.
- Line 262. Here, only the safety of candidate vaccines was evaluated after the assessment of immune indicators. Animal models of Zika virus infection and challenge tests were not established to further verify the efficacy of the candidate vaccines.
Answer: Yes, we evaluated an overview of the immune response promoted by the yeast-based vaccines and some markers that help to check toxicity. In general, the study shows the feasibility of using P. pastoris as a vaccine platform, using an antigen derived from ZIKV as a model. To attest to the efficiency and protection induced by vaccines against virus infection, more robust immunological assays need to be performed.
- Line 349. As the conclusion of the paper, the authors do not seem to summarize the results, specifically which of the two vaccines had the better immune effects.
Answer: We updated the conclusions, highlighting P. pastoris:ENV as the most promising vaccine candidate based on the analyzes performed.

Round 2
Reviewer 1 Report
The authors have answered my questions and comments adequately.
Author Response
Dear Referee 1
Once again, we would like to thank the referee for his work, which allowed for a substantial improvement of our manuscript.
Reviewer 2 Report
The results about humoral immunity were too poor, and IgG titer were a bit low. So I suggest "humoral" should be deleted in title.
The English of the manuscript should be improved.
Author Response
Responses to Reviewer 2
Comment: The results about humoral immunity were too poor, and IgG titer were a bit low. So I suggest "humoral" should be deleted in title.
Answer: We accepted the suggestion and made the change in the title, as well as another revision in the text.
The English in the text has been revised again.
Once again, we would like to thank the referee for his work, which allowed for a substantial improvement of our manuscript.
